# Tree Crown Segmentation and Diameter at Breast Height Prediction Based on BlendMask in Unmanned Aerial Vehicle Imagery

**Jie Xu [1], Minbin Su [1], Yuxuan Sun [1], Wenbin Pan [2], Hongchuan Cui [1], Shuo Jin [1], Li Zhang [1,\*] and Pei Wang [1]**

1   College of Science, Beijing Forestry University, Beijing 100083, China; xu_jie@bjfu.edu.cn (J.X.);
    shanlinsu@bjfu.edu.cn (M.S.); sunyuxuan@bjfu.edu.cn (Y.S.); HongchuanCui@bjfu.edu.cn (H.C.);
    jsdhr666@bjfu.edu.cn (S.J.); wangpei@bjfu.edu.cn (P.W.)
2   School of Humanities and Social Science, Beijing Forestry University, Beijing 100083, China;
    wenpin@bjfu.edu.cn
\*   Correspondence: zhang_li@bjfu.edu.cn; Tel.: +86-010-6233-8136

**Abstract:** The surveying of forestry resources has recently shifted toward precision and real-time monitoring. This study utilized the BlendMask algorithm for accurately outlining tree crowns and introduced a Bayesian neural network to create a model linking individual tree crown size with diameter at breast height (DBH). BlendMask accurately outlines tree crown shapes and contours, outperforming traditional watershed algorithms in segmentation accuracy while preserving edge details across different scales. Subsequently, the Bayesian neural network constructs a model predicting DBH from the measured crown area, providing essential data for managing forest resources and conducting biodiversity research. Evaluation metrics like precision rate, recall rate, F1-score, and mAP index comprehensively assess the method's performance regarding tree density. BlendMask demonstrated higher accuracy at 0.893 compared to the traditional watershed algorithm's 0.721 accuracy based on experimental results. Importantly, BlendMask effectively handles over-segmentation problems while preserving edge details across different scales. Moreover, adjusting parameters during execution allows for flexibility in achieving diverse image segmentation effects. This study addresses image segmentation challenges and builds a model linking crown area to DBH using the BlendMask algorithm and a Bayesian neural network. The average discrepancies between calculated and measured DBH for *Ginkgo biloba*, *Pinus tabuliformis*, and *Populus nigra varitalica* were 0.15 cm, 0.29 cm, and 0.49cm, respectively, all within the acceptable forestry error margin of 1 cm. BlendMask, besides its effectiveness in crown segmentation, proves useful for various vegetation classification tasks like broad-leaved forests, coniferous forests, and grasslands. With abundant training data and ongoing parameter adjustments, BlendMask attains improved classification accuracy. This new approach shows great potential for real-world use, offering crucial data for managing forest resources, biodiversity research, and related fields, aiding decision-making processes.

**Keywords:** BlendMask algorithm; individual-tree crown area–DBH model; Bayesian neural network; image segmentation

## 1. Introduction

### 1.1. Research Significance and Background

Forests, being the cornerstone of terrestrial ecosystems, sustain life for humans and diverse organisms [1], encompassing economic, environmental, social, and cultural values. Yet, human activities continue to diminish forest coverage and resources, resulting in issues like land use changes, ecosystem fragmentation, and biodiversity loss [2]. Comprehensive forest resource surveys serve as critical foundations for effective management strategies and conservation policies, imperative for fostering sustainable forest development. Therefore, at

present, these surveys hold paramount importance for scientific management, sustainable utilization, and protection of forest resources.

Diameter at Breast Height (DBH) is a crucial factor in studying tree structures, and a pivotal variable for forest growth models and management strategies [3]. DBH aids in assessing tree niches, growth status, estimating forest biomass and productivity, monitoring forest health, determining forest structure and species composition, providing fundamental data, and decision-making references for scientific forest resource management and conservation [4].

Forest ecosystems, among Earth's most complex, play a pivotal role in maintaining global biodiversity and ecological balance. DBH data serve as indicators of tree growth and health status. Analyzing DBH data allows for the assessment of forest ecosystem stability, health, and vegetation dynamics. Moreover, DBH is a crucial parameter for estimating tree volume. Evaluating DBH distribution in forests enables calculations of total wood quantity and specific wood levels within a forest area, significantly impacting wood resource assessment, wood industry planning, and economic evaluation [5]. Before advanced remote sensing and digital techniques, forestry resource surveys primarily relied on field investigations and sampling. However, these methods were expensive and lacked precision and comprehensive data. In contrast, remote sensing provides significant benefits by enabling a more precise and efficient collection of extensive forest resource information. Using advanced remote sensing and artificial intelligence algorithms [6], UAV technology allows for high-resolution, multi-temporal, and dynamic monitoring of forest resources. This contributes to an improved evaluation and management of these resources.

With continual advancements in remote sensing technology and the widespread utilization of large-scale remote sensing images [7], remote sensing images now facilitate DBH detection. Remote sensing technology provides a comprehensive, large-scale, and non-destructive means of data acquisition, capturing DBH spatial distribution patterns. Using remote sensing technology for DBH detection enhances work efficiency while reducing human resource demands.

In recent years, groundbreaking advancements in machine learning have found extensive applications across various domains, such as industry, medicine [8], and finance. To address the need for the precise and real-time detection of forestry resources [9], the exploration of machine learning techniques in tree parameter extraction has started. However, this field is in its early stages due to the complexity and variety of algorithms, providing many research possibilities.

In conclusion, using machine learning for extracting tree parameters shows significant potential for forestry resource surveys. This study uses UAV remote sensing images from mixed-tree forests and the BlendMask segmentation model to identify individual trees and measure crown widths. The method includes extracting the contour to calculate the crown area. Finally, a Bayesian neural network creates a model to predict DBH based on measured crown areas in UAV remote sensing images.

*1.2. Research Landscape*

1.2.1. Research Status of Crown Width Extraction

Around 2004, means of digital aerial photogrammetry began to be used for extracting tree crowns [10]. High-resolution remote sensing images were used as the data source, utilizing image processing and computer vision techniques to derive crown shape and contour information. Common approaches included threshold-based segmentation [11], edge-based detection [12], and pixel-based classification [13], albeit limited in precision.

By the early 2010s, the introduction of UAV technology facilitated the acquisition of high-resolution image data for crown extraction [14]. Researchers increasingly employed unmanned aerial vehicles (UAVs) due to their flexibility and maneuverability, enabling a more precise capturing of intricate tree details.

In 2016, Guo Yushan [15] and colleagues used high-spacial-resolution imagery to find tree crowns in both sparse and dense forest areas. They applied the marker-controlled watershed segmentation method, using image gradients to improve the accuracy of extracting crown outlines. Their tests showed an extraction accuracy of 87.8% for sparse forests and 65.5% for dense forests.

Internationally, researchers have conducted studies on crown width extraction, focusing on several key areas:

1. Lidar-based Crown Width Extraction [16]: Lidar technology allows for the highly precise collection of three-dimensional information about ground and canopy surfaces. It is widely used in crown width extraction. Various algorithms, including altitude-threshold-based [17], topological-relation-based [18], and morphological-operation-based [19] approaches, analyze laser point cloud data to extract tree crown information.
2. Image-processing-based Crown Width Extraction [20]: This method uses remote sensing images to extract crown width. By analyzing color, texture, and shape attributes within remote sensing images, the automatic extraction of crown width is achieved.
3. Machine-learning-based Crown Width Extraction [21]: Recent advancements in machine learning algorithms have led to increased exploration of these methods for crown width extraction. Researchers create training sample sets and utilize supervised learning algorithms such as vector machines [22] and random forests [23] to enable the automatic detection and segmentation of crowns.

These studies are beyond extracting crown width and instead focus on exploring the interconnections between crown width and various forest structure parameters. This involves examining how crown width correlates with factors like tree height, diameter at breast height (DBH), tree density, and their impact on forest ecosystem functionality and biodiversity. These insights are useful for managing forest resources and evaluating ecological environments. Both domestic and international research efforts have not only enriched crown width extraction methodologies but also broadened the applications of these data in forest resource research.

To enhance crown width extraction accuracy, researchers continually optimize and improve algorithms and models. They integrate multiple data sources, such as merging lidar data with remote sensing imagery, to obtain comprehensive and accurate information about tree crowns [24]. The increasing use of deep learning methods like convolutional neural networks aids in precisely identifying and segmenting tree crowns. Crown extraction technology is widely used in forest resource management, ecological conservation, climate change research, and related fields. Accurate crown information acquisition facilitates forest structure and biodiversity assessment and helps in monitoring forest health and decision-making and resource management processes.

### 1.2.2. Research Status of Deep Learning in Forestry Segmentation

As deep learning evolves, several models are now used for image recognition, classification, and localization, where convolutional neural networks (CNNs) excel in analyzing two-dimensional images. In addition, CNNs can work with 3D data when the input is converted into a regular form. In 2020, Brage et al. conducted extensive forest surveys using high-resolution satellite imagery. Their use of the Mask R-CNN algorithm for tree crown detection and segmentation achieved precision, recall, and F1-scores of 0.91, 0.81, and 0.86, respectively. This method shows promise in aiding forest resource surveys, planning, and execution. Another study in 2021 by Huang Xinxi focused on ginkgo trees, establishing a dataset of individual ginkgo tree crowns through UAV remote sensing imagery. Utilizing Mask R-CNN and orthophoto maps, they detected tree crowns in different urban settings, achieving a precision rate of 93.90%, a recall rate of 89.53%, an F1-score of 91.66%, and an average precision of 90.86%. Similarly, Huang Yanxiao et al. in 2021 used drones to

capture orthographic images of two distinct Metasequoia forest plots. They improved the Faster R-CNN method for crown identification and width extraction, resulting in a more accurate model with 92.92% accuracy and a determination coefficient of 0.84, showing enhancements over the original model.

BlendMask uses a Mask R-CNN-based framework that merges object detection and instance segmentation. During object detection, BlendMask uses two simultaneous branches: one for creating the object's bounding box and another for generating a rough segmentation mask. In the segmentation phase, BlendMask selectively extracts the target using the bounding box network and aligns feature maps of various scales to a fixed size via ROIAlign. Subsequently, the fusion mask generator refines feature maps into precise segmentation masks using a sequence of convolution operations, with each layer integrating a fusion module to enhance feature expression. Ultimately, BlendMask utilizes the predicted bounding box and segmentation mask to produce detection and segmentation outcomes. Presently, there is limited documentation or practical use of the BlendMask model in the forestry segmentation domain, presenting substantial research opportunities in this field.

### 1.2.3. Research Status of DBH Prediction of Trees

In forestry resource estimation, individual-tree diameter at breast height (DBH) is a critical evaluation metric. Yet, field surveys have long grappled with challenges like high difficulty and slow-paced data collection. Traditional forestry surveys often rely on empirically derived formulas relating crown area to DBH to estimate actual tree diameter. However, in reality, different tree species exhibit significant variations in crown area and DBH, posing difficulties in establishing correlations [25]. In recent years, the surge in machine learning achievements across various domains has drawn widespread attention, gradually expanding applications in forestry. Models based on neural networks offer new insights into understanding the relationship between tree crown area and DBH.

Accurate DBH data play a crucial role in forestry resource surveys. For instance, Fu Kaiting used an EBEE unmanned aerial vehicle equipped with a digital camera to capture high-resolution images within the Tanli management area at the Nanning Arboretum in Guangxi [26]. Following orthographic image processing, they established a DBH–crown width regression model. This model efficiently correlated individual tree crown width with measured DBH, enabling a swift estimation of stand volume by integrating individual tree volume models [27]. Similarly, Shi Jieqing and collaborators developed an all-encompassing forest resource survey system using UAV remote sensing images, integrating various UAV photography techniques and post-processing technologies alongside GIS. Impressively, reported errors in tree number density and volume were merely 2.68% and 4.01%, respectively, promoting forest resource exploration [28].

Tree DBH prediction often relies on traditional regression models in international studies. Methods such as linear regression, multiple linear regression [29], and generalized linear models [30] are frequently used to model and predict the relationship between DBH and other dependent variables, such as tree height, age, and growth environment. As machine learning technologies evolve, an increasing number of researchers explore employing machine learning algorithms for tree DBH prediction. Remote sensing data [31] also significantly influence DBH prediction. By using high-resolution remote sensing images and lidar data from forest areas, relevant geographical, morphological, and structural features can be extracted and used as input variables for constructing and optimizing tree DBH prediction models.

To enhance the accuracy of DBH prediction, researchers also attempt to fuse different types of data. Integrating multiple data sources, such as ground survey data, remote sensing data, and environmental factors [32], can reveal the factors affecting DBH more comprehensively and improve the accuracy and stability of prediction models. In recent years, deep learning methods have seen increased application in tree DBH prediction.

Convolutional neural networks (CNNs) [33] and recurrent neural networks (RNNs) [34] can extract image features and time series features, culminating in precise DBH predictions.

*1.3. Primary Research Focus*

In crown segmentation techniques, the threshold-based segmentation algorithm is a common method. It separates images based on preset thresholds applied to pixel grayscale values. However, it can be inaccurate in complex backgrounds, varying lighting, and occlusion situations. It struggles especially with irregular or overlapping crown shapes [35].

Another approach involves using features and a classifier for segmentation. This method extracts texture, shape, and color features from images, and then uses a classifier to label pixels belonging to the crown region. Yet, accurately segmenting crowns in complex backgrounds remains challenging due to limitations in feature selection and classifier design [36]. Additionally, manual parameter adjustments are often needed for different tree species and environments.

The third method turns crown segmentation into an image segmentation problem using region-based or edge-based algorithms. However, these methods can be computationally complex, with large datasets and complex scenes [37], and may result in false segmentation with overlapping trees and unclear boundaries. The fourth method, multi-sensor fusion-based segmentation, combines data sources like remote sensing images and lidar data to provide more comprehensive information for crown segmentation. But, challenges persist in data fusion and registration technologies, requiring sophisticated multi-sensor data collection and processing [38].

To improve the computational efficiency and increase the recognition accuracy, we adopted a more streamlined and efficient model: the BlendMask model [39] for crown segmentation.

In the calculation of tree DBH, linear regression equations are widely used in the industry, but they often result in poor fitting effects. In this study, we opted for a Bayesian neural network to model the relationship curve between crown area and DBH. Combining the application of convolutional neural networks in aerial remote sensing images, we devised a precise individual-tree crown segmentation method based on high-resolution UAV images and the BlendMask network. This methodology enabled the extraction of crown information from trees within the forest farm. The specific research objectives include:

1. Utilizing the orthophoto map of Beijing Jingyue Ecological Forest Farm as experimental data to use the BlendMask network for segmenting individual crowns and detecting the count of Pinus tabulaeformis trees.
2. Assessing the prediction results of the model using relevant accuracy evaluation metrics.
3. Fitting an optimal relationship model between the DBH and crown width of trees using a Bayesian neural network, leveraging DBH measurements of sample trees collected in the field and the calculated crown mask area obtained from segmentation.

**2. Research Area and Data Acquisition**

*2.1. Field Investigation and Data Acquisition*

2.1.1. Research Area

The research area of this project is located near the forest farm of Jingyue Ecological Forest Farm, Baishan Town, Changping District, Beijing. The geographic coordinates are 116°19′7.2192″E and 40°11′32.002″N (Figure 1). The area has a warm temperate semi-humid continental monsoon climate, with an annual mean temperature of about 11.8 °C and an annual mean precipitation of 550.3 mm. The soil is yellow loam, suitable for the growth of warm temperate broad-leaved forest and coniferous forest.

There are eight tree species in the study area, including three coniferous and five broad-leaved species. The different shapes of tree crowns in the forest farm pose higher challenges for the applicability and performance of the crown segmentation system.

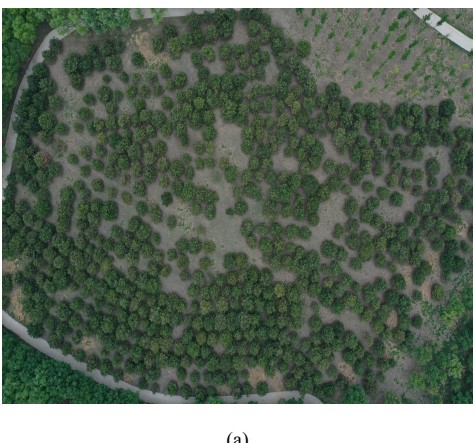
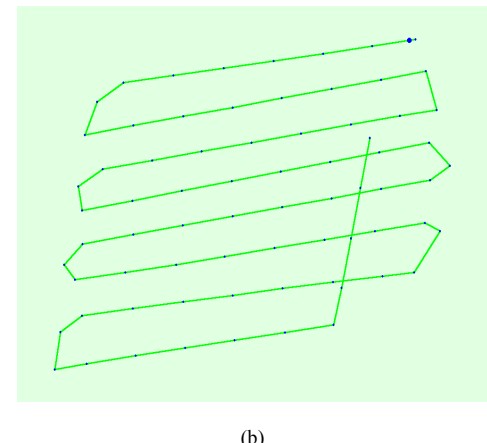

(a)

(b)

**Figure 1.** (**a**) Aerial photo of forest farm. (**b**) Top view of the initial image position (the green line starts from the big blue dot and follows the image position in time)

2.1.2. Field Investigation and Data Collection

On 20 June 2023, we conducted a field survey to delineate the study area's boundaries, document various tree species, and determine suitable data collection methods. We formulated a comprehensive survey plan, specifying the UAV flight route and height and assessing potential flight and safety risks. From 8 to 10 July 2023, the members of the experimental group visited the forest farm to collect relevant information about the designated tree species. On the morning of 11 July 2023, when the weather in the study area was clear and the wind speed was low, we captured a set of aerial images along the predetermined route using a UAV after a pre-flight test. The UAV model used for the aerial photography was the DJI MAVIC 2 PRO (The manufacturer of this product is DJ-Innovations, located in Shenzhen, China), which had a 1-inch CMOS sensor, a 28 mm focal length lens with a 77° viewing angle, a maximum photo resolution of 5472 × 3648 pixels, and a maximum flight time of about 31 min.

The UAV flew at a speed of 4 m/s and a height of 50 m while taking aerial photographs. Photos were captured every 5 s, resulting in a total of 355 images that fully covered the necessary area. The image resolution was 5472 × 3648 pixels, and each pixel represented an actual area of 1.46 units.

After collecting the images, all the trees in the sample plot were numbered and DBH values were measured. Considering that the relationship between the crown area of a single tree and its diameter at breast height are different for different tree species, the study was modeled using split-sample plots. Therefore, sample plots were set up from forest areas of the same age and the same species in sections with a distance of 6 m or more from the forest edge and with normal growth of trees in the stand. The sample plot boundaries were determined with a measuring rope.

At a distance of 1.3 m from the base of each tree, a diameter measuring tape was placed completely around the trunk and pressed against the surface of the trunk. The corresponding value of the breast diameter was read, which is the tree's DBH. To ensure the reliability of the data, additional measurements and calculations were made to accurately measure the diameter at breast height for trunks with significant leaning or shapes.

*2.2. Datasets Creation*

2.2.1. Synthesis of Orthophoto Map

In this study, the original UAV images underwent preprocessing using Pix4dmapper v4.4.12 software to create an orthographic image of the chosen plot. The images were imported into Pix4dmapper, and a three-dimensional digital surface model (DSM) of the entire research area was generated through three key steps: sparse point cloud reconstruction, densification of the point cloud, and texture mapping.

The first step involved reconstructing a sparse point cloud by extracting key features from the UAV images. This process involved identifying and matching common points between overlapping images to estimate their 3D coordinates. Next, the sparse point cloud was refined to add more detail to the terrain and objects in the study area. This step interpolated additional points within the sparse point cloud to increase its density and capture finer landscape details. Finally, the high-resolution orthographic image obtained from the dense point cloud was mapped onto the surface to provide accurate texture information. This process aligned the orthographic image with the reconstructed 3D model and projected it onto corresponding surfaces.

By following these steps in Pix4dMapper software, a realistic three-dimensional representation of the research area was generated (Figure 2), which facilitated further analysis and interpretation of the crown segmentation and tree count detection tasks.

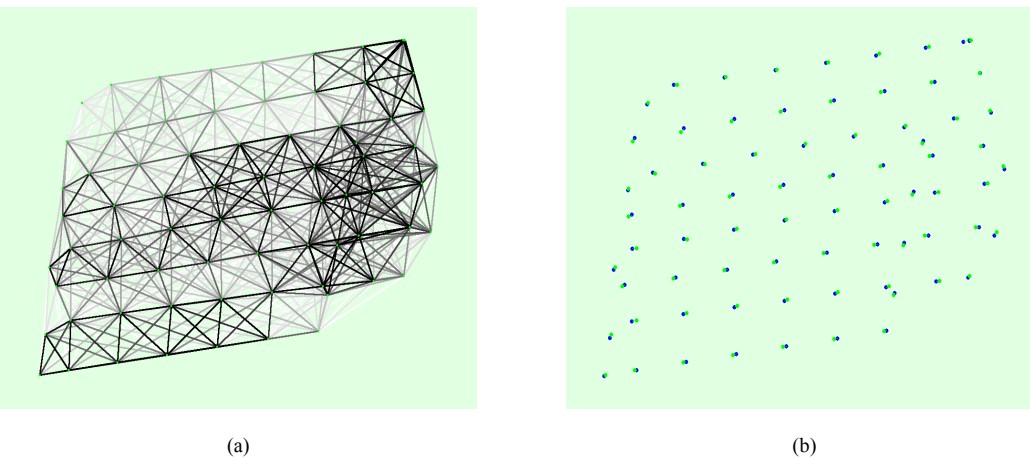

(a)                                                                 (b)

**Figure 2.** (**a**) Depicts a top-view illustration presenting the image positions computed through connections among matched images. Dark links represent the count of matched 2D keypoints between the digital images, while brighter links signify weaker matches that may necessitate manual point connections or additional images. (**b**) The discrepancies between the initial positions (blue dots) and the calculated positions (green dots) of the images are showcased, along with the differences between the initial positions (blue crosses) and the calculated positions (green crosses) of the ground control points (GCPs) in the top (XY plane), front (XZ plane), and side (YZ plane) views.

We used the forward mapping function of Pix4dMapper to produce an orthophoto (DOM) of the area from the three-dimensional model (Figure 3) to prevent data loss due to insufficient processor memory and the structure and functional parameters are shown in Table 1.

**Table 1.** Relevant parameter values set by Pix4dMapper during 3D modeling of real scene.

| Step | Type | Specific Settings |
|---|---|---|
| **Densification of point cloud** | Image scale | 1/2 |
| | Point cloud density | best |
| | Minimum matching number | 3 |
| | Matching window size | 7 × 7 pixels |
| **Three-dimensional grid** | Configuration | Medium resolution |
| | Sampling density distribution | 1 |
| **Texture mapping** | Texture color source | Visible color |
| | Texture compression quality | 75% JPEG image quality |
| | Maximum texture size | 8192 |
| | Texture sharpening | Enabled |

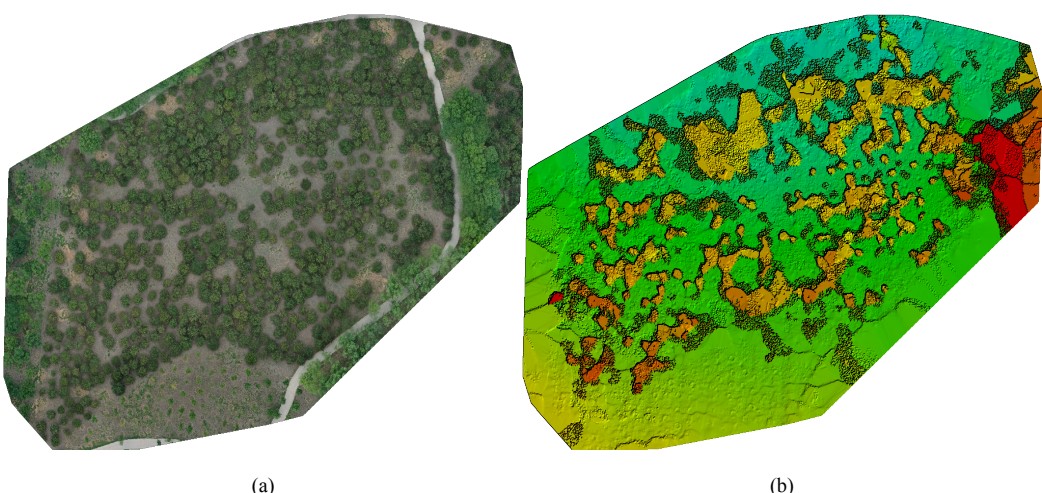

(a)                                                                              (b)

**Figure 3.** (**a**,**b**) show the orthogonal surface model before densification and corresponding sparse digital surface model (DSM).

### 2.2.2. Generating Label Samples

The large orthophoto image of the study area (Figure 4) was carefully sliced into 820 images, each sized at 1024 × 1024 pixels using Photoshop v7.0 software. Augmentation techniques, including rotation, flipping, brightness adjustments, and noise enhancement, were applied to diversify the dataset.

For the creation of a specific dataset targeting the segmentation of individual tree crowns, the Labelme v3.16.2 annotation software was used. Pinus tabulaeformis were the primary subject for algorithmic training. The crowns of these trees within the experimental area were manually outlined using the polygon labeling feature in Labelme. To minimize potential errors, both field survey data and visual interpretation were combined during the annotation process.

Subsequently, the dataset was divided into three subsets: a training set, a validation set, and a test set. The training set comprised 752 images, the validation set contained 210 images, and the test set encompassed 238 images. These subsets were utilized for training, evaluating, and testing the performance of the crown segmentation algorithm, respectively.

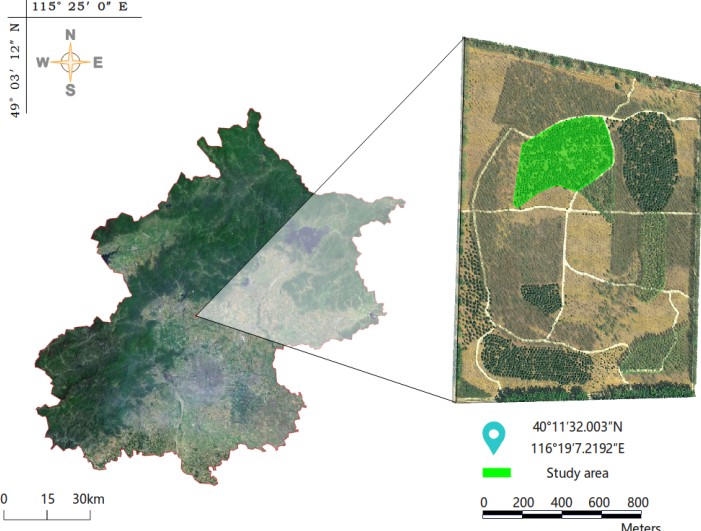

**Figure 4.** General information of the study area. (**Left**) Geographical location of the study area. (**Right**) The remote sensing image of the study plots.

*2.3. Evaluation Metrics*

2.3.1. Accuracy Assessment of Individual Tree Detection

In evaluating the precision of individual tree counting, we will employ metrics such as precision, recall, and F1-score [40].

$R_{Precision}$ measures the accuracy of this model in instance segmentation:

$$R_{\text{Precision}} = \frac{n_{TP}}{n_{TP} + n_{FP}} \times 100\% \tag{1}$$

where true positives ($n_{TP}$) indicate the number of correctly predicted positive samples while false positives ($n_{FP}$) indicate the number of samples incorrectly predicted as positive by the model.

$R_{recall}$ assesses the accuracy of the model's recognition:

$$R_{\text{recall}} = \frac{n_{TP}}{n_{TP} + n_{FN}} \times 100\% \tag{2}$$

where false negatives ($n_{FN}$) represent the number of samples wrongly predicted as negative cases by the model.

F1-score, a harmonic average, provides a comprehensive evaluation combining precision and recall:

$$F1 - \text{score} = 2 \times \frac{R_{\text{Precision}} \times R_{\text{recall}}}{R_{\text{Precision}} + R_{\text{recall}}} \times 100\% \tag{3}$$

2.3.2. Crown Segmentation Accuracy Metrics

For crown segmentation accuracy, we will utilize standard evaluation metrics [41]—average precision (AP) and mean average precision (mAP)—for prediction frames and instance segmentation masks.

Average precision (AP) gauges model accuracy under varying confidence thresholds:

$$AP = \sum (R_{\text{recall}}[i] - R_{\text{recall}}[i-1]) \times R_{\text{Precision}}[i] \tag{4}$$

In the formula, $R_{recall}[i]$ indicates the recall rate under the i-th confidence threshold and $R_{Precision}[i]$ indicates the accuracy under the i-th confidence threshold.

Mean average precision (mAP) is a metric obtained by averaging multiple average precisions:

$$mAP = \frac{\sum_{i=1}^{n} AP[i]}{n} \tag{5}$$

where $\sum_{i=1}^{n} AP[i]$ represents the average precision of the i-th category and n represents the number of categories.

2.3.3. Individual Tree Crown Area and DBH Accuracy Metrics

For the crown area and DBH, we will adopt the measurement accuracy evaluation metrics commonly used for continuous values, including relative error (Re), mean absolute error (MAE), and root mean square error (RMSE) [42].

Relative error (RE) measures the difference between the predicted value and the actual value:

$$RE = \frac{\widehat{y_l} - y_i}{y_i} \tag{6}$$

Mean absolute error (MAE) measures the average error between the predicted value and the actual value:

$$MAE = \frac{1}{n} \sum_{i=1}^{n} |\hat{y}_l - y_i| \tag{7}$$

Root mean square error (RMSE) measures the average error between the predicted value and the actual value:

$$RMSE = \sqrt{\frac{1}{n}\sum_{i=1}^{n}(\hat{y}_i - y_i)^2} \tag{8}$$

In the above categories, $\hat{y}_i$ means the predicted value and $y_i$ means the actual value.

## 3. Research Methods

### *3.1. Crown Segmentation Method*

3.1.1. Watershed Algorithm

The watershed algorithm is a commonly used traditional technique for segmenting tree crowns. It uses geographical morphology to mimic structures like mountains, valleys, basins, and trees for categorization. Geodesic distance is a crucial concept in this method [43].

By using the watershed algorithm with geodesic distance, the algorithm can effectively identify and segment individual tree crowns from the surrounding background. This method enhances accuracy in distinguishing tree crowns, providing valuable information for further analysis and measurements.

In Figure 5, the algorithm approximates the tree crown as an ellipse. The Euclidean distance between two black points is represented by the length of the dashed line segment (d45), while the geodesic distance denotes the shortest actual path distance, ideally the minimum sum of distances along the real path [44]; that is, $d_{12} + d_{23} + d_{34} + d_{45}$ .

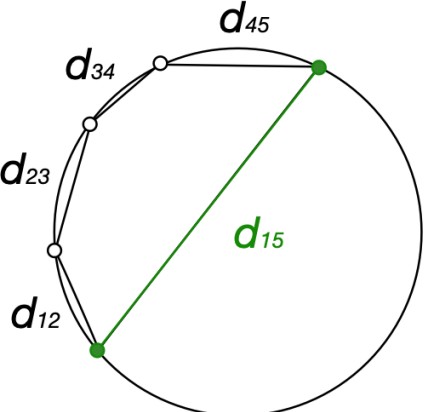

Minimum Sum = $d_{12} + d_{23} + d_{34} + d_{45}$

**Figure 5.** Watershed algorithm shows the column.

This algorithm partitions adjacent pixels with close grayscale values into regions. The process of estimating the diameter at breast height (DBH) using the watershed algorithm is outlined as follows:

1.  Convert tree crown images in the dataset to grayscale and classify pixels based on grayscale values, establishing a geodesic distance threshold.
2.  Identify the pixel with the minimum grayscale value (defaulted as the lowest) and incrementally increase the threshold from the minimum value, designating these as starting points.
3.  As the plane expands horizontally, it interacts with neighboring pixels, measuring their geodesic distance from the starting point (lowest grayscale). Pixels with distances below the threshold are submerged, while others have dams set, thus categorizing neighboring pixels.

4.  Use the complementary canopy height model (CHM) distance transform image for segmentation. Utilize the h-minima transform to suppress values smaller than 'H', generating a marker image for tree tops, followed by reconstruction through erosion.

Each tree is then marked using the region minimum method, where the 'H' threshold directly influences tree top identification. Finally, the watershed algorithm segments the image based on the extracted tree top markers.

### 3.1.2. BlendMask Algorithm

BlendMask is a model used for instance segmentation that extends the Mask R-CNN framework. It combines modifications and enhancements to Mask R-CNN to improve the accuracy and efficiency of instance segmentation.

BlendMask leverages the advantages of Mask R-CNN and introduces an attention network and a feature blending module to facilitate the segmentation process. The feature blending module is responsible for fusing features from different scales to extract richer semantic information. It consists of a three-level feature pyramid network (FPN), where each level has an adaptive pooling layer to capture features at various scales. Compared to the blurry boundaries of Mask R-CNN, BlendMask is highly sensitive to boundaries and can achieve fine-grained object segmentation. It has shown outstanding performance in tasks such as edge detection and image segmentation.

Remarkably, no researcher has applied the BlendMask model to the precise segmentation of individual tree crowns in complex environments. Moreover, the BlendMask model adopts some optimization techniques, such as the auto-augmentation [45] and joint segmentation scheme, to further improve its performance.

Based on these considerations, this study selects the BlendMask algorithm to segment tree crowns and compares it with the conventional watershed algorithm. By evaluating the performance of both algorithms, we can gain insights into the effectiveness and suitability of BlendMask for accurately segmenting tree crowns in complex environments.

### 3.2. Calculation of Crown Area

In this study, the dataset images undergo processing with OpenCV for filtering and conversion to grayscale. The findContours method is a function in the OpenCV library that is commonly used to find and analyze the contours of objects in binary images. It is utilized to identify the contours within the images. Then, the drawContours method is applied to display the canopy and rectangular outlines on the original images, helping to verify contour accuracy.

To calculate the actual area of the marked rectangle in the image and determine the true crown area, the ratio of pixels occupied by the rectangle to those within the crown region is computed using the formula:

$$S_i = S_r \times \frac{P_i}{P_r} \tag{9}$$

where $S_i$ represents the real area of the crown, $S_r$ represents the real area of the rectangle, $P_i$ represents the pixel value of the crown area, and $P_r$ represents the pixel value of the rectangle. When calculated, the actual area per pixel is 1.46 cm$^2$.

### 3.3. Crown Area–DBH Model

In forest resource estimation, the crown area and DBH of individual trees are critical metrics. However, field surveys often encounter difficulties such as complexity and sluggish data collection speed. Moreover, substantial variations in crown area and DBH among different tree species pose challenges in establishing correlations [46].

Recent advancements in machine learning have garnered widespread attention across various domains, including forestry. Its increasing application in forestry research offers novel avenues for exploring the relationship between crown area and DBH.

Traditional backpropagation (BP) neural networks use multiple fully connected layers for data fitting, often resulting in overfitting and poor generalization. Consequently, these models may lack confidence in their predicted outcomes [47].

In contrast, the Bayesian neural network (BNN) integrates Bayesian algorithms for optimized convolution. Its structural design (shown in Figure 6) combines probabilistic modeling with neural network techniques. It treats weight parameters as probabilities rather than fixed values. Prior distributions describe critical parameters, serving as inputs, and the network's output characterizes the probability distribution likelihood. The posterior distribution is computed using sampling or variational inference. Despite slower training, Bayesian neural networks offer strong generalization and provide confidence estimates for predictions, making them suitable for predictive modeling.

Thus, this study chooses the Bayesian neural network to model the relationship between crown area and DBH. It compares its performance with the traditional BP neural network to understand the effectiveness and suitability of using Bayesian neural networks for predicting the crown area–DBH relationship.

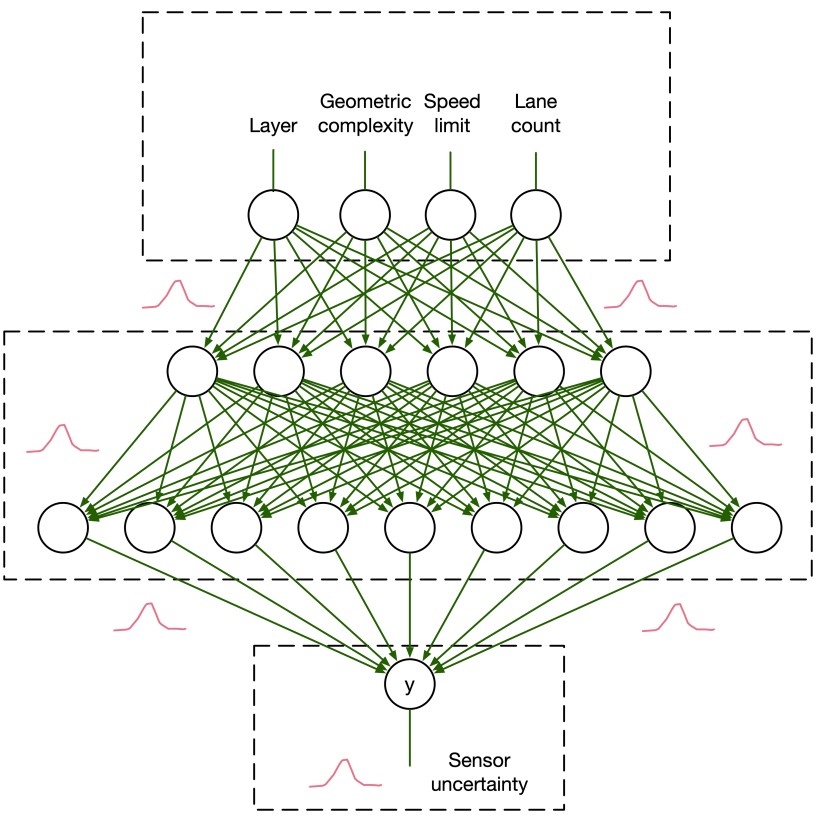

**Figure 6.** BNN structure.

## 4. Research Results

### 4.1. Model Training

The adjustment in model network training parameters significantly impacts the overall model's training and predictive efficacy. BlendMask uses the stochastic gradient descent algorithm (SGD) for training. In this experiment, the BatchSize is set to 4 and the initial learning rate is adjusted to 0.0035, with the training process spanning 100 epochs and 200 iterations per epoch. The hardware attributes of the workstation are shown in Table 2 which is configured with Intel e5-2640 central processing unit (The manufacturer is INTEL, located in Santa Clara, USA), NVIDIA RTX 6000 graphics processor unit (The manufacturer is NVIDIA, located in Santa Clara, USA), 1T solid-state drive (The manufacturer is SAMSUNG, located in Gyeonggi-do, Republic of Korea).

**Table 2.** Workstation hardware attribute.

| Hardware | Attribute |
|----------|-----------|
| CPU | E5-12640 |
| GPU | RTX 6000 24GB |
| SSD | 1T SSD |
| Memory | 32GB |

### *4.2. Data Processing and Preprocessing*

#### 4.2.1. Crown Segmentation of Individual Tree

For the individual-tree crown segmentation adopted in this study (shown as Figure 7), preprocessed samples and annotated images are fed into the BlendMask network for training. Model loss is determined post-training, with model parameter updates facilitated through gradient backtracking until achieving optimal parameterization. Finally, the optimized model is deployed for the precise detection of individual tree crown edges, accurate segmentation, and total count within the entire aerial orthophoto image.

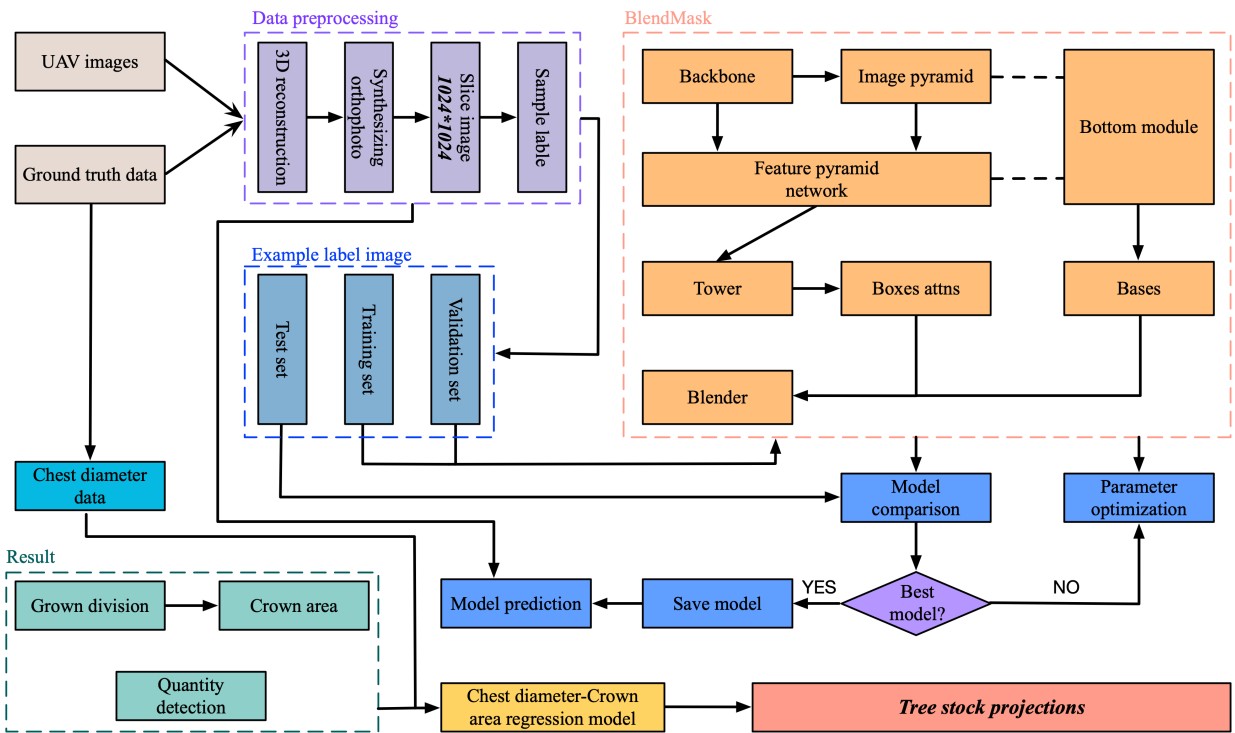

**Figure 7.** Process of individual tree crown segmentation and DBH prediction.

#### 4.2.2. Testing the Individual Tree Crown Area–DBH Model

The computed crown area and measured DBH data are presented in Table 3. This research selects 70% of the effective samples as training data to establish the correlation function curve between crown area and DBH. Additionally, 15% of the samples are allocated for model accuracy verification while the remaining 15% are reserved for model testing purposes.

**Table 3.** Data of crown area and DBH of samples.

| Total Sample Number | Mean DBH (cm) | Maximum Breast Diameter (cm) | Minimum DBH (cm) | Average Crown Area (m²) | Maximum Crown Area (m²) | Minimum Crown Area (m²) |
|---------------------|---------------|------------------------------|------------------|-------------------------|-------------------------|-------------------------|
| 164 | 15.5892 | 22.5781 | 9.5670 | 6.8217 | 12.1171 | 2.5853 |

### 4.3. Evaluation of BlendMask's Performance in Individual Tree Crown Segmentation

#### 4.3.1. Crown Segmentation Effect

The combination of top-down and bottom-up approaches in BlendMask effectively integrates features from both the bottom layer and the top layer. This fusion enables the model to capture detailed information and boundary characteristics of tree crowns, resulting in more precise crown segmentation results. Moreover, it ensures that there is no overlap between the segmented crown boundaries.

Figure 8a–c are the original images.

Figure 8d,g,j illustrate the performance of BlendMask in segmenting *Pinus tabulaeformis* crowns. The model demonstrates the ability to accurately capture the intricate details of the crown edges without missing any parts or generating false detections. The overall segmentation and edge detection effect for *Pinus tabulaeformis* crowns is excellent.

Figure 8e,f,h,i,k,l showcase the application of BlendMask in the identification of *Ginkgo biloba* and *Populus nigra varitalica* trees. It is notable that this algorithm also achieves remarkable results for these tree species. It effectively captures the distinctive features of their crown widths, demonstrating accurate segmentation and excellent edge detection capabilities.

Through its combination of top-down and bottom-up methods, BlendMask exhibits enhanced performance in capturing detailed crown information, ensuring precise segmentation results, and effectively identifying various tree species.

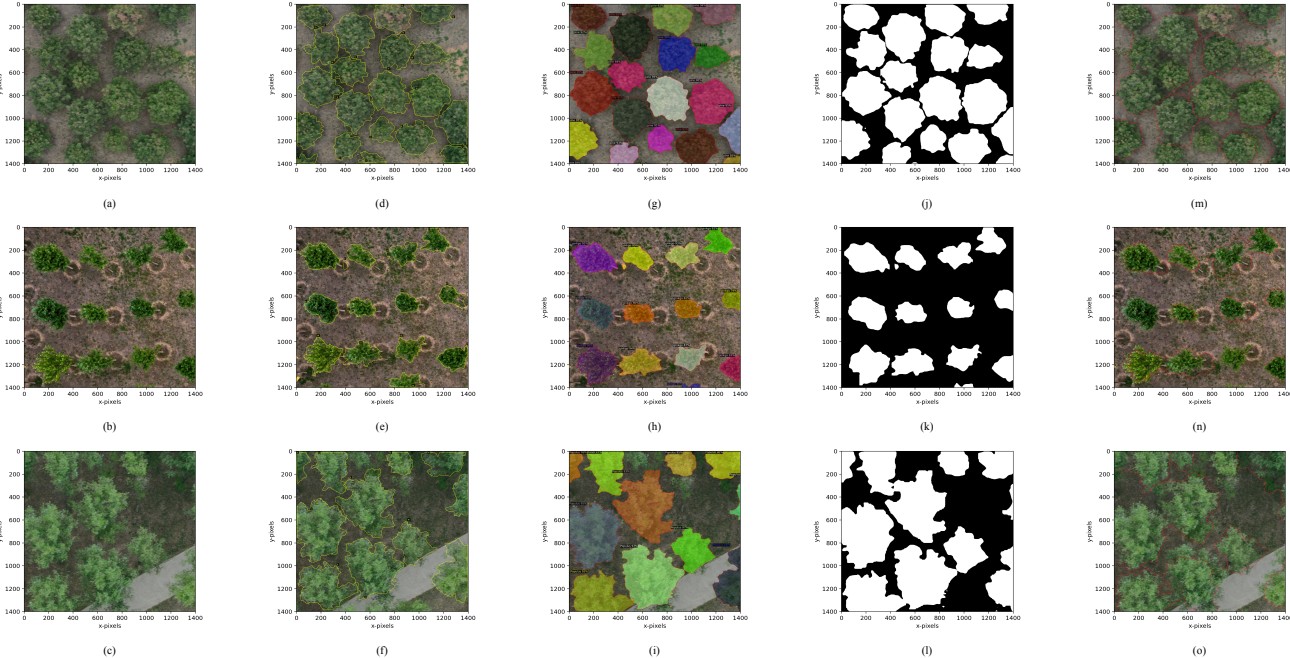

**Figure 8.** Output result of BlendMask and watershed single-tree crown segmentation.

Figure 8m–o are the output images when using the watershed algorithm for crown segmentation. From Figure 8m–o, it is clear that while the watershed algorithm can accurately identify and extract most crowns, there are still instances of merging, misclassification, and omission in the results, indicating limitations in achieving precise crown segmentation.

However, the BlendMask algorithm in the same figure shows superior segmentation performance compared to the watershed algorithm. It effectively deals with merging, misclassification, and omission issues encountered in the watershed algorithm. This results in more accurate and reliable segmentation, improving tree crown identification and extraction.

The comparison between the BlendMask and watershed algorithms clearly indicates the advantages of the BlendMask algorithm in achieving superior crown segmentation outcomes. Its ability to capture detailed information, handle complex boundaries, and overcome the limitations of traditional methods makes it a more effective approach for crown segmentation tasks.

4.3.2. Evaluation of Crown Segmentation Performance

The average precision and mean average precision represent the ratio of overlap between the crown width obtained through BlendMask segmentation and the actual crown width. Higher values indicate better model performance.

Subsequently, the crowns are segmented using the watershed method. The actual crown distribution is initially divided into five groups, with four groups as training datasets and one as a verification dataset. This process repeats five times, using each of the five groups once. Parameters like morphological structural element (B) and H in the h-minima transform vary between 0.5, 1.0, 1.5, 2.0 and 3, 5, 7, 9, respectively. The accuracy rate ACI of crown classification is presented in Table 4:

Comparing Table 5, representing the overlap ratio between the crown widths obtained by BlendMask segmentation and the actual crown widths, the minimum average precision in Table 5 is 0.724, while the maximum is 0.893 at IOU = 0.5. The maximum value for the correct rate ACI in Table 4 from the watershed segmentation is 0.721, obtained when H = 1.5 and B = 3.

**Table 4.** Correct rate of crown segmentation ACI.

| / | h = 0.5 | h = 1.0 | h = 1.5 | h = 2.0 |
|---|---------|---------|---------|---------|
| b=3 | 0.696 | 0.715 | 0.721 | 0.707 |
| b=5 | 0.675 | 0.689 | 0.695 | 0.686 |
| b=7 | 0.661 | 0.647 | 0.653 | 0.656 |
| b=9 | 0.624 | 0.612 | 0.625 | 0.611 |

**Table 5.** Evaluation index of crown segmentation performance.

| Model Weight | Average Precision | Mean Average Precision IOU = 0.5 | Mean Average Precision IOU = 0.75 |
|--------------|-------------------|----------------------------------|-----------------------------------|
| BlendMask | 0.724 | 0.893 | 0.745 |
| Watershed algorithm | 0.685 | 0.763 | 0.674 |

In summary, the BlendMask algorithm in this study outperforms the watershed algorithm in crown segmentation performance.

4.4. *Performance Evaluation of Crown Area–DBH Model*

The BlendMask algorithm computes the crown area, utilizing measured DBH values of trees as training samples for both traditional BP neural networks and Bayesian neural networks. The training involves refining these models iteratively until reaching the desired performance, followed by testing using different data. Table 6 showcases the calculation accuracy indices for the crown area and DBH of individual trees. On the other hand, Table 7 provides the network training parameters used in the training process.

To verify the fitting performance of the two BP neural networks for crown area and DBH, evaluation indices are presented in Table 8. The fitting results and error distribution can be observed in Figure 9, while the regression situation is illustrated in Figures 10 and 11. These tables and figures provide a comprehensive assessment of the performance and

accuracy of the trained models. They assess the effectiveness of BP neural networks in predicting crown area and DBH based on BlendMask-calculated crown areas. They offer insights into how well the models capture the relationship between crown area and DBH, enabling accurate predictions and measurements of these tree attributes.

Figure 9 clearly demonstrates that the goodness of fit for the fitting models of both neural networks is greater than 0.9. This indicates that the fitting models of both the traditional BP neural network and the Bayesian neural network effectively capture and reflect the relationship between crown area and DBH. However, comparing training and test set performances reveals differences. The traditional BP neural network has a higher training set performance but considerably lower test set performance, indicating overfitting. In contrast, the Bayesian neural network demonstrates robust generalization capabilities, providing superior predictive results beyond the training dataset.

In summary, the Bayesian neural network outperforms the traditional BP neural network in modeling the relationship between crown area and DBH. It not only predicts accurately within the dataset but also exhibits stronger generalization abilities, making it more proficient at predicting crown area and DBH values beyond the dataset's limits.

**Table 6.** Calculation accuracy index of crown area and DBH of individual tree.

| / | Relative Error RE | Average Absolute Error MAE | Root Mean Square Error RMSE |
|---|---|---|---|
| **Crown area** | 0.05653 | 0.3290 | 0.4563 |
| **DBH** | 0.03308 | 92.18 | 106.4 |

**Table 7.** Network training parameters.

| Target Training Times | Learning Rate | Minimum Error of Training Target | Additional Momentum Factor | Minimum Performance Gradient |
|---|---|---|---|---|
| 10000 | 0.001 | 0.000001 | 0.95 | 0.00001 |

**Table 8.** Evaluation indicators of different models.

| Model | Training Set/$R^2$ | Test Set/$R^2$ | All/$R^2$ | RMSE |
|---|---|---|---|---|
| **Traditional BP neural network** | 0.96523 | 0.90999 | 0.9456 | 0.74516 |
| **Bayesian neural network** | 0.9488 | 0.95628 | 0.94775 | 0.72602 |

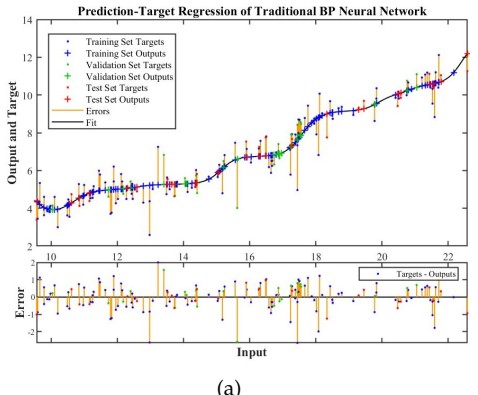 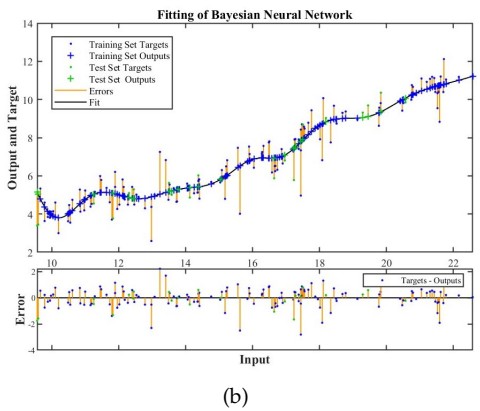

(a)  (b)

**Figure 9.** (**a**) Fitting of Traditional BP Neural Networktwo neural networks. (**b**) Fitting of Bayesian Neural Networks.

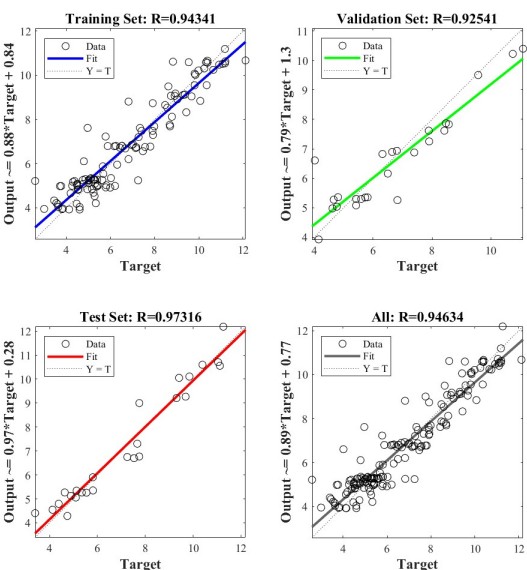

**Figure 10.** Prediction target regression of traditional BP neural network.

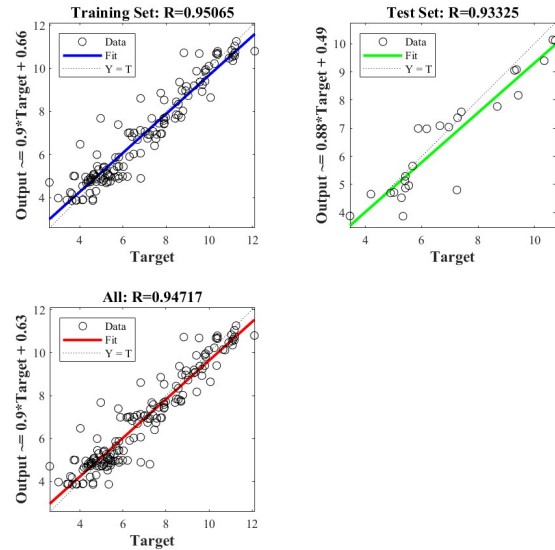

**Figure 11.** Prediction target regression of Bayesian neural network.

## 4.5. Comparative Analysis

Comparing the data in Table 9. As previously discussed, the BlendMask algorithm outperforms the watershed algorithm in crown segmentation, while the Bayesian neural network excels over the traditional BP neural network in fitting crown area to DBH relationships. However, their combined effects might differ in crown width extraction and breast diameter prediction. Therefore, we conducted a comparison of four combined methods and present the experimental outcomes below:

The results conclusively demonstrate that the combination of the BlendMask algorithm for crown segmentation and the Bayesian neural network for DBH prediction produces the most optimal outcomes. Specifically, the BlendMask algorithm effectively segments tree crown widths, allowing for the precise extraction of crown contours. The subsequent use of the Bayesian neural network establishes a strong correlation between crown area and DBH, resulting in accurate DBH predictions. This integrated approach significantly enhances the precision and consistency of tree identification and measurement by providing more accurate crown segmentation and DBH measurement values.

**Table 9.** Comparative evaluation of algorithm combination.

| Model | Training Set | Test Set | All |
|---|---|---|---|
| **BlendMask** <br> **+ Bayesian neural network** | 0.7855 | 0.7926 | 0.7862 |
| **BlendMask** <br> **+ raditional BP neural network** | 0.69882 | 0.65883 | 0.6846 |
| **BlendMask** <br> **+ raditional BP neural network** | 0.69882 | 0.65883 | 0.6846 |
| **Watershed** <br> **+ traditional BP neural network** | 0.6499 | 0.6551 | 0.6492 |

*4.6. Final Function Validation*

In this study, field data collection and validation were conducted across a range of tree species. Utilizing our model's formula correlating crown area with DBH, we estimated the crown areas of *Pinus tabuliformis*, *Ginkgo biloba*, and *Populus nigra varitalica*. These estimations enabled DBH calculations for each species, which were then compared against actual measurements. The findings, detailed in Table 10, revealed that the average discrepancies between calculated and measured DBH for *Ginkgo biloba*, *Pinus tabuliformis*, and *Populus nigra varitalica* were 0.15 cm, 0.29 cm, and 0.49cm, respectively, all within the acceptable forestry error margin of 1 cm.

To verify the model's efficacy, extensive repeated trials were performed on all specimens of *Pinus tabuliformis*, *Ginkgo biloba*, and *Populus nigra varitalica* within the forest. These trials consistently demonstrated that both the crown area and DBH of each tree closely approximated actual measurements, indicating robust model performance. However, due to constraints in manuscript length, only a subset of representative trees from the sampled plots were selected for detailed statistical analysis. The results from these samples indicated average DBH errors of 0.11 cm for *Pinus tabuliformis*, 0.28 cm for *Ginkgo biloba*, and 0.31 cm for *Populus nigra varitalica*, further corroborating the model's precision.

**Table 10.** Comparison between measured diameter and calculated diameter of tree at breast height.

| Sample Area | The Tree Number | MDTBH/cm | CDTBH/cm | DTBH Error/cm |
|---|---|---|---|---|
| *Pinus tabulaeformis* | No.8 | 9.56 | 9.43 | 0.13 |
| | No.15 | 9.98 | 9.94 | 0.04 |
| | No.29 | 10.1 | 10.20 | 0.10 |
| | No.41 | 10.84 | 10.72 | 0.12 |
| | No.53 | 11.79 | 11.92 | 0.13 |
| *Ginkgo biloba* | No.13 | 17.28 | 17.41 | 0.13 |
| | No.18 | 18.53 | 18.67 | 0.14 |
| | No.25 | 18.81 | 18.69 | 0.12 |
| | No.31 | 19.32 | 19.23 | 0.09 |
| | No.44 | 19.45 | 19.25 | 0.20 |
| *Populus nigra varitalica* | No.4 | 23.32 | 23.42 | 0.10 |
| | No.11 | 23.81 | 23.78 | 0.03 |
| | No.19 | 24.54 | 24.85 | 0.31 |
| | No.23 | 24.98 | 24.72 | 0.26 |
| | No.29 | 25.10 | 25.01 | 0.09 |

## 5. Discussion

*5.1. Comparative Experimentation under Varying Light Intensities*

Environmental factors significantly affect crown segmentation in the study area. For example, the leaf color of the same tree species may differ due to various environmental conditions such as light intensity. Choosing the best time for image capture can notably improve recognition accuracy [48]. To evaluate the BlendMask model's applicability in

UAV-based aerial photography for single-tree crown segmentation under diverse lighting conditions, this study simulated different light intensities by adjusting UAV image brightness within a range of 50% to 150%, starting from an initial illumination of 100%. Figure 12 depicts segmentation outcomes at brightness levels of 50%, 75%, 100%, 125%, and 150%.

The results reveal a gradual improvement in predictive performance with increasing light intensity. At lower light intensities, such as 50%, the BlendMask model exhibits uneven edge segmentation (seen in Figure 12b). For high prediction accuracy, an illumination range of approximately 75% to 150% is recommended. Hence, capturing crown images during periods of strong light intensity, like noon, enables the model to generate precise predictions. This strategy also aids accurate calculations of crown area and DBH predictions.

Understanding the impact of light intensity on segmentation and selecting optimal shooting conditions improves the BlendMask model's reliability and performance in aerial photography involving individual tree species captured by UAVs.

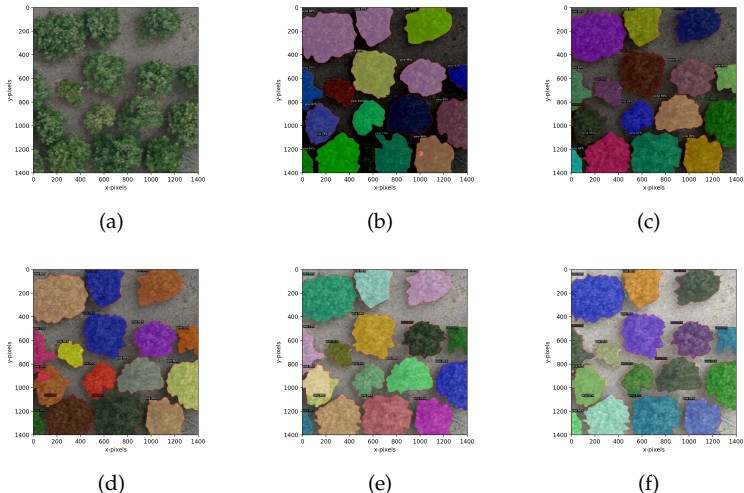

**Figure 12.** (**a**) Original image (**b**) Predictions at 40% brightness (**c**) Predictions at 70% brightness (**d**) Predictions at 100% brightness (**e**) Predictions at 130% brightness (**f**) Predictions at 160% brightness.

### 5.2. Analysis of Incorrect Segmentation Cases

The BlendMask model makes mistakes in segmentation, especially in dense tree areas or where trees have unusual crown shapes. For instance, Figure 13b shows an example where closely spaced tree trunks led to overlapping masks created by the BlendMask algorithm. Occasionally, multiple masks were generated for the same tree. This issue might stem from challenges encountered during the three-dimensional reconstruction and orthophoto synthesis stages. If the alignment of feature points used in model training is inaccurate, it hampers precise mask delineation, causing difficulties in distinguishing closely positioned tree trunks.

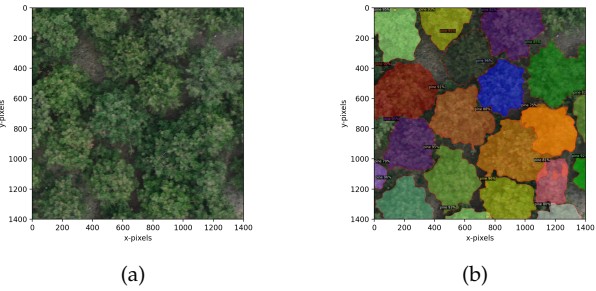

**Figure 13.** (**a**) Original image; (**b**) crown segmentation results and error cases of BlendMask.

*5.3. Image Segmentation Analysis of Different Sizes*

In practical scenarios, image sizes often differ. Hence, the model's ability to handle images of various sizes is crucial. In this study, larger-sized images were tested, as shown in Figure 14: the BlendMask-created masks effectively outlined crown edges when dealing with image sizes of 2000 × 2000 and 3000 × 3000, demonstrating the model's proficiency in handling images of different dimensions.

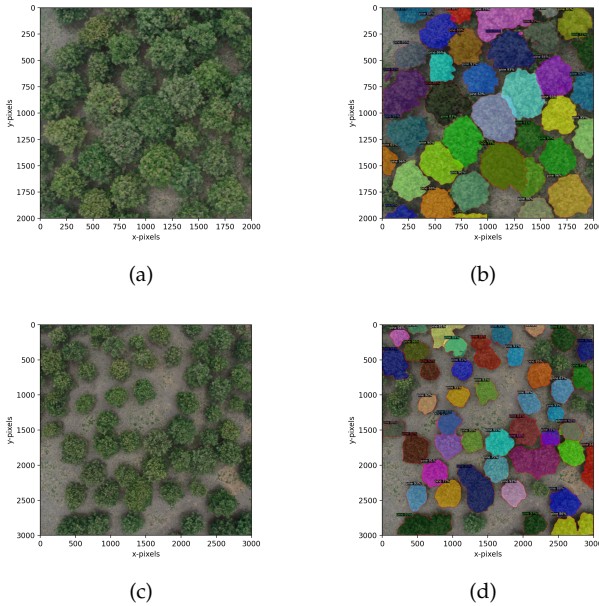

**Figure 14.** (**a**) Original image with image size of 2000 × 2000. (**b**) The crown segmentation result of BlendMask in (**a**). (**c**) The original image with the image size of 3000 × 3000. (**d**) The crown segmentation result of BlendMask in (**c**) .

## 6. Conclusions

This study used the BlendMask algorithm for individual tree crown segmentation and integrated a Bayesian neural network to predict diameter at breast height (DBH) based on crown area measurements. The method addressed over-segmentation issues in the watershed algorithm while maintaining detailed crown edge information across various scales. Flexible parameter adjustments during segmentation allowed for diverse image segmentation effects.

Compared to traditional algorithms, this study brings forth several notable advantages:

1. BlendMask's Multi-step Approach: BlendMask utilizes a two-step method for instance segmentation in complex scenes. BlendMask initially extracts the region of interest (ROI) using a pretrained target detector and then performs segmentation of the ROI. Integrating deep learning models, BlendMask delivers more accurate and precise outcomes in handling complex segmentation tasks.

2. Robustness to Obstructions and Overlaps: BlendMask effectively handles challenges related to occlusions and overlaps using deep learning models, particularly when distinct objects within a scene overlap or obscure each other. This robustness was beneficial in training Pinus tabulaeformis stand crown information, especially in cases of occluded and intertwined crowns.

3. Scalability: BlendMask's adaptability to large-scale datasets enhances segmentation performance by extracting richer features. It can be applied to various vegetation datasets, aiding in identifying diverse tree crown shapes, sizes, and distributions. This contributes to a comprehensive understanding of forest spatial structures, ecological attributes, and growth patterns.

Specifically, the BlendMask algorithm can process and analyze forest aerial or remote sensing images, extracting the crown outline and positional data. When combined with lidar or 3D scanners, it provides three-dimensional crown information (height, volume, shape), offering precise descriptions of crown details [49].

**Author Contributions:** Conceptualization, L.Z.; methodology, J.X.; software, M.S.; validation, J.X. and Y.S.; formal analysis, J.X.; investigation, J.X., M.S. and Y.S.; resources, L.Z.; data curation, H.C.; writing—original draft preparation, J.X., W.P. and S.J.; writing—review and editing, L.Z.; visualization, J.X.; supervision, L.Z. and P.W.; project administration, L.Z.; funding acquisition, L.Z. All authors have read and agreed to the published version of the manuscript.

**Funding:** This research was funded by Beijing Municipal Natural Science Foundation, grant number No. 6232031.

**Data Availability Statement:** The datasets used and/or analysed during the current study are available from the corresponding author on reasonable request.

**Acknowledgments:** We are very grateful to all the students assisted with data collection and the experiments. We also thank anonymous reviewers for helpful comments and suggestions to this paper.

**Conflicts of Interest:** The authors declare no conflicts of interest.

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
