# Peer review of "Tree Crown Segmentation and Diameter at Breast Height Prediction Based on BlendMask in Unmanned Aerial Vehicle Imagery"

_remotesensing, doi:10.3390/rs16020368_

Round 1

Reviewer 1 Report

Comments and Suggestions for Authors

Introduction gives otherwise good review on the scientific background of this study, but perhaps more references could be included concerning the general use of machine learning in the remote sensing of forests.

Data processing is well described concerning the remote sensing data and photogrammetric workflow.

In the accuracy assessment, the presentation of the accuracy metrics seems incomplete (see my further comments below).

Field reference data is very superficially described and lacking specific details.

The methodology is mostly described in an appropriate way, but many of the variables used in the equations have not been explained (see chapter 2.3.).

Presentation of results is appropriate.

The linguistic form is good and understandable.

l. 45 Usually DBH is used in tree level models, thus it would be better to talk about models estimating tree volumes (instead of wood volume).

l. 61-65 There are already quite a lot of experiments carried out on machine learning, and the authors could search some general references here on this topic.

l. 76 The time period in correct although there are a number of experiments before that, see e.g. https://silvafennica.fi/show/mono/mono3.

l. 123 CNNs also work with 3D data, if the input is made to regular form.

l. 239-243 The collection of field data is described very superficially.

Since this is an empirical study based on real field material, a more detailed description is needed, e.g.: what is the unit of measurement (tree/stand?), number of observations, what variables were recorded in the field?

In chapter 2.3. Not all applied symbols and abbreviations are listed. These should be added.

l. 385 Is there a reference for the findContours method?

l. 391-392 Something seems to be missing in this sentence:

“Where i represents the real area of crown, r represents the real area of rectangle, represents the pixel value of crown area and represents the pixel value of rectangle.”

In figure 8, there  are sub-figures (e.g. m, n & o) to which I could not find any mention or reference in the figure caption or in the main text. What do they stand for and why are they included?

The list of references is appropriate, although highly emphasizing national/domestic research work.

Author Response

Dear reviewer,
Thank you very much for taking the time to review this manuscript. Please find the detailed responses below and the corresponding revisions/corrections highlighted in the re-submitted files.
Comments 0: Introduction gives otherwise good review on the scientific background of this study, but perhaps more references could be included concerning the general use of machine learning in the remote sensing of forests. 
Response 0: Thanks for your valuable feedback. We have checked the literature carefully and added more reference on machine learning in our revised manuscript and highlighted in the paper.
Comments 1: Data processing is well described concerning the remote sensing data and photogrammetric workflow.
Response 1: We feel great thanks for your valuable feedback.
Comments 2: In the accuracy assessment, the presentation of the accuracy metrics seems incomplete (see my further comments below).

Field reference data is very superficially described and lacking specific details.
Response 2: Thank you for your suggestion and we have added more details in chapter 2.1.2. (l. 256-267)
Comments 3: The methodology is mostly described in an appropriate way, but many of the variables used in the equations have not been explained (see chapter 2.3.). 
Response 3: We sincerely appreciate the valuable comments. We have re-writen this part according to the your suggestion and highlighted in our revised manuscript.(l.305-339)
Comments 4: Presentation of results is appropriate. 

The linguistic form is good and understandable.
Response 4: We feel great thanks for your valuable feedback.
Comments 5: l. 45 Usually DBH is used in tree level models, thus it would be better to talk about models estimating tree volumes (instead of wood volume).
Response 5: We sincerely thank the reviewer for careful reading.As suggested by the reviewer,we have corrected the “wood” into “tree”. Thanks for your correction.(l. 45)

Comments 6: l. 61-65 There are already quite a lot of experiments carried out on machine learning, and the authors could search some general references here on this topic.

Response 6: Thank you for your suggestion and we have added two references on the topic.(l. 61-63)

Comments 7: l. 76 The time period in correct although there are a number of experiments before that, see e.g. https://silvafennica.fi/show/mono/mono3.

Response 7: Thanks for your suggestion and we have added this earlier experiment and its reference in this section. “Around 2004,means of Digital Aerial Photogrammetry began to be used for extracting tree crowns. ”(l. 75-76)

Comments 8: l. 123 CNNs also work with 3D data, if the input is made to regular form.

Response 8: Thank you for pointing this out. We agree with this comment. Therefore, we have added ‘In addition, CNNs can work with 3D data when the input is made to regular form.’ in this section.(l. 125-126)

Comments 9: l. 239-243 The collection of field data is described very superficially.

Since this is an empirical study based on real field material, a more detailed description is needed, e.g.: what is the unit of measurement (tree/stand?), number of observations, what variables were recorded in the field?

Response 9: Thank you for your suggestion and we have added a description of field data collection in chapter 2.1.2. (l. 256-267)

Comments 10:In chapter 2.3. Not all applied symbols and abbreviations are listed. These should be added.

Response 10: Thanks for your correction and we have modified chapter2.3 and added all applied symbols and abbreviations.(chapter 2.3)

Comments 11: l. 385 Is there a reference for the findContours method?

Response 11: We feel great thanks for your professional review work on our article and we have tried our best to add explanations of the findContours method in the revised manuscript.(l. 398-400)

Comments 12: l. 391-392 Something seems to be missing in this sentence:

“Where i represents the real area of crown, r represents the real area of rectangle, represents the pixel value of crown area and represents the pixel value of rectangle.”

Response 12: We feel sorry for our carelessness. In our resubmitted manuscript we have modified this sentence.“Where Si represents the real area of crown, Sr represents the real area of rectangle, Pi represents the pixel value of crown area and Pr represents the pixel value of rectangle.”(l. 406-407)

Comments 13: In figure 8, there are sub-figures (e.g. m, n & o) to which I could not find any mention or reference in the figure caption or in the main text. What do they stand for and why are they included?

Response 13: We feel sorry for our carelessness. Figure 8 (m,n,o) are the output images when using the watershed algorithm for crown segmentation. We have made changes and additions in the revised manuscript.(l. 474-475)

We would like to express our great appreciation to you for comments on our paper.

Thank you and beat regards.

Sincerely,

Xu Jie

Reviewer 2 Report

Comments and Suggestions for Authors

The paper study an interesting topic from an extremely current point-of-view. It is in well-written and all the faced issues are supported by a large bibliography. The changes have improved the readability of the paper even if there are some minor remarks that should be taken into consideration.

Section 1.1 –This is only a suggestion. More recent literature could be added to better reflect  the great significance of the research and its wide range of applications.

Section 2.1 – I think the section lacks a description of the source of the data "DBH".Are seedlings being measured? What kind of tool was used to measure it? The method of measurement needs more clear explanation.

Section 4.5 –It is unclear why this section only has 4.5.1. Section4.5.1 is a general evaluation of the combined model. That's why I think this part should be separated from section4.5.

Typos:

Figure 4 –In the caption,"of the study" appears twice .I think this is a typo that must be corrected.

Figure 12 –Wrong capital letter in “Image”.

Figure 13 –The word "Orthophoto" is misspelled.

Comments on the Quality of English Language

None.

Author Response

Dear reviewer,

Thank you very much for taking the time to review this manuscript. Please find the detailed responses below and the corresponding corrections highlighted in the re-submitted files.

Comments 0: The paper study an interesting topic from an extremely current point-of-view. It is in well-written and all the faced issues are supported by a large bibliography. The changes have improved the readability of the paper even if there are some minor remarks that should be taken into consideration. 

Response 0: We feel great thanks for your valuable feedback that we have used to improve the quality of our manuscript.

Comments 1: Section 1.1 –This is only a suggestion. More recent literature could be added to better reflect  the great significance of the research and its wide range of applications.

Response 1: We feel great thanks for your valuable feedback and according to your nice suggestion,we have added more references in this section,(l. 52, 55-56)

Comments 2: Section 2.1 – I think the section lacks a description of the source of the data "DBH".Are seedlings being measured? What kind of tool was used to measure it? The method of measurement needs more clear explanation.

Response 2: Thank you for your suggestion and we have added a description of field data collection and more details in chapter 2.1.2.(l. 256-267)

Comments 3: Section 4.5 –It is unclear why this section only has 4.5.1. Section4.5.1 is a general evaluation of the combined model. That's why I think this part should be separated from section4.5.

Response 3: We sincerely appreciate the valuable comments. We have modified this part according to the your suggestion and highlighted in our revised manuscript.(l. 548)

Comments 4: Typos: 

Figure 4 –In the caption,"of the study" appears twice .I think this is a typo that must be corrected.

Response 4: We have corrected this sentence.

Comments 5: Figure 12 –Wrong capital letter in “Image”.

Response 5: We sincerely thank the reviewer for careful reading.As suggested by you,we have corrected the “Image” into “image”. Thanks for your correction.(chapter5.1)

Comments 6: Figure 13 –The word "Orthophoto" is misspelled.

Response 6: We feel sorry for our carelessness. In our resubmitted manuscript we have corrected this word into “Original image”.(chapter5.2)

Thank you and beat regards.

Sincerely,

Xu Jie
